# Exploring the physical, mental, and social dimensions of middle-aged adults for active and healthy aging: A cross-sectional study

Eunice Santos[1,2,3☯*], Lara Guedes Pinho[2,4☯¤], Adelaide Proença[5,6☯], Helena Arco[5,6☯]

1 Institute for Research and Advanced Training, Universidade de Évora, Évora, Portugal, 2 Comprehensive Health Research Centre, CHRC, LA-REAL, Universidade de Évora, Évora, Portugal, 3 Unidade Local de Saúde do Baixo Alentejo, Beja, Portugal, 4 Nursing Department, Universidade de Évora, Évora, Portugal, 5 Polytechnic Institute of Portalegre, Portalegre, Portugal, 6 CARE—Research Center on Health and Social Sciences, Polytechnic Institute of Portalegre, Portalegre, Portugal

☯ These authors contributed equally to this work.
¤Current address: University of Évora, Évora, Portugal
* d52903@alunos.uevora.pt

## Abstract

### Introduction

Population ageing presents a significant global challenge, necessitating sustained efforts to promote active and healthy ageing throughout life to improve quality of life in later years. This study aims to characterise the physical, mental, and social well-being of middle-aged adults (aged 55–64) in Baixo Alentejo, Portugal, and to analyse associations between these dimensions and sociodemographic variables. The findings aim to inform policies and interventions supporting active and healthy ageing, a cornerstone for quality longevity.

### Methodology

This cross-sectional, descriptive study was conducted between 02 May 2023 and 29 February 2024 among individuals aged 55–64 registered at health centres in Baixo Alentejo, Portugal. Data were collected via a structured questionnaire evaluating disability, depressive symptoms, life satisfaction, and satisfaction with social support. Instruments included the WHO Disability Assessment Schedule (WHODAS 2.0-PT12), the Patient Health Questionnaire (PHQ-9), a self-reported life satisfaction score, and the Social Support Satisfaction Scale (SSSS). Statistical analysis employed Student's t-test and one-way ANOVA. Ethical approval was obtained, and all participants provided informed consent.

### Results

The study included 698 participants. Women, individuals with lower educational attainment, and the unemployed demonstrated significantly higher functional disability scores. Women and unemployed participants also had higher depressive symptom scores. Conversely, men reported greater life satisfaction. Older participants and those with lower socioeconomic status exhibited greater physical limitations, depressive symptoms, and

**Data availability statement:** All relevant data are within the manuscript.

**Funding:** The author(s) received no specific funding for this work.

dissatisfaction with social support. Economic stability was positively associated with mental well-being and life satisfaction, underscoring the importance of financial security in enhancing perceptions of social support.

## Conclusion

This study provides a comprehensive characterisation of middle-aged adults in Baixo Alentejo, revealing significant associations between sociodemographic factors and physical, mental, and social well-being. The findings highlight the need for tailored socio-economic and health interventions to promote active and healthy ageing. Public policies designed to address the unique needs of middle-aged adults in the region are critical to improving health outcomes and fostering quality longevity.

## Introduction

Population ageing is a global phenomenon that presents significant social and health challenges, necessitating substantial transformations in healthcare systems worldwide [1,2]. Portugal, one of Europe's most rapidly ageing nations, faces pronounced challenges, particularly in regions like Baixo Alentejo. This region exhibits one of the highest ageing rates alongside substantial population decline, underscoring the urgent need for regional strategies to foster active and healthy ageing among its middle-aged population [3–6].

The Alentejo region, which includes the Baixo Alentejo, could see a significant decline in population between 2000 and 2050, with the resident population falling by 46.1 per cent [1]. In particular, the young population is expected to fall by 70.4 per cent and the number of pre-school children by 72.9 per cent [1]. This makes it the Portuguese region with the highest ageing rate (189.2 per cent) [1–3].

Baixo Alentejo (NUTS III) is a predominantly rural region with the largest surface area in the country (8,544.6 km²) and the lowest population density (14.7 inhabitants/km²) [3,4,7]. Demographic challenges are exacerbated by long distances between towns, which can exceed 120 km, and a limited and inefficient public transport network [3,5,6,8]. Geographical dispersion and transport barriers make it difficult to provide effective health care, especially for the most dependent [2,5,8]. In this context, the ageing of the population in the Baixo Alentejo is a critical challenge, highlighting the urgent need for strategies to mitigate the socio-economic and public health impact of this phenomenon [2,3,9].

Targeted interventions for individuals aged 55–64—a life stage where functional and socioeconomic changes begin to emerge—are crucial for mitigating the risks of disability and promoting quality ageing. Evidence from longitudinal studies, such as the Survey of Health, Ageing and Retirement in Europe (SHARE), demonstrates the potential of tailored interventions during this critical period to significantly reduce disability rates and enhance quality of life in later years [10–12]. Middle age marks the onset of health, functional, and socioeconomic transitions that directly influence ageing trajectories, autonomy, and independence [8,13].

The ageing process is inherently multifaceted, driven by biological, psychological, social, and contextual determinants. These interrelated factors influence the development of health issues, functional disabilities, and dependency, ultimately impacting quality of life [14–19]. Addressing these complexities at an earlier stage of the ageing continuum is vital for reversing negative trends. The concept of active ageing, which encompasses autonomy, independence, and capacity-building, has emerged as a key framework for promoting healthspan and quality

of life [16,20]. Active ageing involves lifelong learning [21], participation in societal activities, and the use of technology to empower individuals throughout their lives [22,23].

Existing literature supports a multifactorial model of active ageing, which includes physical, mental, and social dimensions shaped by personal, behavioural, economic, and environmental determinants [15,24,25]. This model highlights the interconnectedness of health, safety, participation, and learning as foundational pillars for quality ageing, as conceptualised by the World Health Organisation (WHO) [12,20,26,27]. Moreover, recent research has drawn attention to the role of structural inequalities, including economic and environmental factors, in shaping active ageing outcomes [20].

Healthy ageing, as defined by the WHO, is characterised by the maintenance of physical, mental, and social functionality, which fosters autonomy and quality of life [8]. It necessitates creating inclusive and supportive environments while preventing disease across the lifespan. This holistic perspective underscores the critical interplay of individual and environmental factors in shaping ageing trajectories, particularly within economically and socially diverse contexts [28–32].

Although global studies on ageing are extensive, there is a lack of integrative research addressing the physical, mental and social well-being (three dimensions simultaneously) of middle-aged adults in Portugal, particularly in the Baixo Alentejo, where demographic changes and socio-economic challenges require targeted attention [3,4,9,33]. Existing studies mainly focus on the elderly population and rarely explore the intersectoral dimensions of well-being among middle-aged adults in this under-researched region [3,4,10,11]. This gap limits the development of policies and interventions tailored to regional needs. Given that middle age represents a pivotal window for preventive action, addressing physical, mental, and social dimensions during this stage is essential for promoting active ageing strategies [13].

This study aims to (i) characterise the physical, mental and social well-being of middle-aged adults (55–64 years old) in Baixo Alentejo, Portugal, and (ii) to analyse the associations between these dimensions and sociodemographic variables. By providing a comprehensive overview, this study seeks to inform the development of sustainable public policies and targeted interventions that promote active and healthy ageing, contributing to quality longevity both within Baixo Alentejo and in similar contexts across Europe.

## Materials and methods

### Study design and setting

An observational, cross-sectional, descriptive, and quantitative study was conducted using a questionnaire-based survey. Following approval from the Baixo Alentejo Local Health Unit (ULSBA), questionnaires were distributed to health centres in the ULSBA region, specifically in Aljustrel, Almodôvar, Alvito, Barrancos, Beja, Castro Verde, Cuba, Ferreira do Alentejo, Mértola, Moura, Ourique, Serpa, and Vidigueira. All regions in the ULSBA catchment area were included in the study. Data collection was carried out between 02 May 2023 and 29 February 2024.

### Population and sample size

The target population included all individuals aged 55–64 registered at ULSBA health centres in 2023, representing a total of 17,786 individuals. This age range corresponds to the classification of middle-aged adults [34]. Baixo Alentejo aligns with the statistical territorial unit NUT III [35]).

The minimum sample size was determined to be 391 participants, based on a 95% confidence level and a 5% margin of error. A total of 704 responses were collected, of which 698

were deemed valid after excluding 6 improperly completed questionnaires. Stratified random sampling was used to ensure representative coverage, with nurses distributing and collecting questionnaires among the selected participants. Stratified random sampling was used to ensure representative coverage, with nurses distributing and collecting questionnaires from selected participants. This method allowed participants to be categorised according to specific demographic factors such as age, gender and socio-economic status, ensuring that all relevant subgroups were adequately represented. Nurses played a crucial role in the data collection process, distributing and collecting questionnaires directly from participants. This approach ensured the systematic inclusion of diverse populations and maintained the integrity of the data by ensuring equal opportunities for participation among eligible individuals.

## Study participants and data collection

Eligible participants were individuals aged 55–64 and registered at ULSBA health centres. Each person attending the health centre for a consultation was invited by the nurse to participate in the study. The nurse explained the aims of the study, and if the person agreed, they were given the questionnaire to complete at that time, with sufficient time allowed. Exclusion criteria included adults with dementia or cognitive deficits, individuals unable to read, understand, or write in Portuguese, and those incapable of providing informed consent due to other limitations. As the questionnaire was written in Portuguese and was self-administered, participants were required to be able to read and understand the language, which was essential for understanding the content and providing accurate responses.

## Addressing potential biases

Potential biases related to data collection methods, including comprehension bias, recall bias, and impulsive responses, were anticipated. Measures to mitigate these included ensuring a comfortable and private response environment and offering opportunities for clarification by the principal investigator (first author) or the nurse responsible for distributing the questionnaire.

## Questionnaire

The questionnaire consisted of two sections. The first section contained questions regarding the socio-demographic characterisation of the participants (age, gender, marital status, educational and academic qualifications, professional situation and socio-economic status according to self-perceived monthly income). The second section consisted of the measurement instruments. The dimensions of physical, mental and social well-being were analysed using a model established by Bosch-Farré et al. (2018) [12]. We complemented it by using a few different scales to measure each dimension. The physical wellbeing dimension was assessed with the WHODAS 2.0 PT-12; the mental wellbeing dimension with the PHQ-9 and the self-perceived life satisfaction score, used in 2018 by Bosch-Farré et al. [24]; the social dimension was approached using the Pais Ribeiro Social Support Satisfaction Scale (2011) [13]. This choice of scales was based on their validity and relevance to the study population.

## Characterisation of WHODAS 2.0 PT-12

The WHODAS 2.0 (World Health Organisation Disability Assessment Schedule) is a tool developed by the World Health Organisation (WHO) to assess an individual's level of functionality and disability [14]. The 'PT-12' version refers to the 12-item questionnaire translated into Portuguese by Moreira et al., 2015 [15].

The 12-item WHODAS 2.0 PT-12 contains 12 Likert scale questions with possible answers ranging from 'none' to an 'extreme' degree depending on the difficulty in a functional area over the last 30 days. It can be self-completed. It assesses six main domains of functionality:

1. Cognition: Understanding and communication.

2. Mobility: Ability to move and be independent.

3. Self-care: Skills related to looking after oneself.

4. Relationships with other people: Interpersonal interactions and relationships.

5. Life activities: Carrying out household chores and leisure activities.

6. Participation: Participation in society, including work and social activities.

The score for the short version (of 12 items) is a simple score because the scores assigned to each of the items 'none' = 1, 'mild' = 2, 'moderate' = 3, 'severe' = 4, 'extreme' = 5 are added together [14]. The WHODAS manual states that the simple score is sufficient to 'describe the degree of functional limitations', due to the scale's unidimensional structure and high internal consistency [14].

The possible scores for the 12-item WHODAS 2.0 range from 12 to 60; a score of 12 indicates that, for all items, the answer was ''no'' degree of difficulty in functionality. In the validation study for the Portuguese population, good internal consistency was obtained (α = 0.86) [15] The questionnaires considered valid were those that were completely answered, with all 12 items answered, or with only one item missing. In this case we applied the simple approach for missing data as indicated in the WHODAS manual [14].

## Characterisation of the Patient Health Questionnaire (PHQ-9)

The Portuguese version of the Patient-Health Questionnaire-9 (PHQ-9), consisting of 9 items, was used to assess the presence of depressive symptoms. It can be self-completed. The PHQ-9 was developed by Kroenke et al. (2001) to promote the recognition and diagnosis of mental illness [3]. It can be scored both categorically, to provide probable diagnoses, and continuously, to provide the level of depressive symptoms [16]. The items are categorised on a 4-point ordinal scale, indicating how often the symptoms have been present in the last two weeks, with 'never = 0', 'several days = 1', 'more than half the number of days = 2' and 'almost every day = 3'. The last question on this scale refers to the level of functional deficit, revealing the respondent's overall perception of the impact these symptoms have on work, looking after things at home or socialising with other people. The two-week period chosen represents the time frame used to diagnose major depression. The total score is obtained by adding up the ratings of all the items on the scale, with 0 as the minimum value and 27 as the maximum value. Values between 0 and 4 indicate no depression, between 5 and 9 mild depression, between 10 and 14 moderate depression, between 15 and 19 moderately severe depression and between 20 and 27 severe depression [17].

A higher PHQ-9 score indicates a higher level of depression. In the validation study for the Portuguese population, the scale showed adequate internal consistency, with an α = .86 [16].

## Characterisation of the Social Support Satisfaction Scale (SSSS)

Pais Ribeiro's Social Support Satisfaction Scale [13] was developed to measure satisfaction with existing social support. It is a measure of perceived social support, on the assumption that this perception is a fundamental dimension in the cognitive and emotional processes linked to well-being and quality of life [13]. It consists of 15 sentences that are presented for self-completion as a set of statements. The participant must tick the degree of agreement with the statement on a Likert scale with five positions, 'totally agree', 'mostly agree', 'neither agree nor

disagree', 'mostly disagree', and 'totally disagree'. The scale is made up of four dimensions. The first dimension, entitled 'satisfaction with friends', assesses contentment with friendships and friends. It consists of five items (items 3, 12, 13, 14, 15) and has an internal consistency of 0.83.

The second dimension, called 'intimacy', measures the perception of intimate social support. This dimension contains four items (items 1, 4, 5, 6) and has an internal consistency of 0.74.

The third dimension, called 'satisfaction with family', analyses satisfaction with the family support available. It includes three items (items 9, 10, 11) and also has an internal consistency of 0.74.

The fourth dimension, referred to as 'social activities', assesses satisfaction with the social activities the person does. This dimension is made up of three items (items 2, 7, 8) and has an internal consistency of 0.64.

The Satisfaction with Social Support Scale (SSSS) offers five scores: one for each dimension and an overall score that results from combining all the dimensions. The score for each dimension is obtained by adding up the corresponding items for each factor, while the total score for the scale is the sum of all the items. The internal consistency of the total scale is 0.85 [13].

The items are evaluated by assigning a value of '1' to answers marked 'A' and '5' to those marked 'E'. The scale includes inverted items, which are: items 4, 5, 9, 10, 11, 12, 13, 14 and 15. For these, the value '1' is assigned to answers marked as 'E' and '5' for those marked as 'A'.

The total score on the scale ranges from 15 to 75, with higher scores indicating a greater perception of social support. There are no cut-off values that can be considered indicative of deficiency; in other words, all individuals have a perception of satisfaction with social support, and a low or high score does not necessarily imply a lack of support.

To determine the score for each dimension, the items that make it up must be added together. As the number of items varies between the dimensions, the minimum and maximum scores also differ. If comparisons are necessary, it can be useful to convert the scores to a scale of '0' to '100', where '0' corresponds to the lowest possible score for the dimension and '100' to the highest. The answers refer to the last 30 days, allowing for a temporal assessment.

## Self-perceived life satisfaction score

Satisfaction with life is a personal, introspective and global opinion about what a person thinks they are and what they have done with their life [18].

Self-reported satisfaction with life was assessed on a scale of 0 to 10 points, where 0 means 'not at all satisfied' and 10 means 'extremely satisfied' with life. In this instrument, participants are asked to position themselves at that moment according to their satisfaction with life. Values equal to or greater than 7 mean satisfaction with life [12].

## Independent variables

In this study we used six sociodemographic variables, which have previously been identified in the literature as relevant to characterising active and healthy ageing in individuals over 50. These variables include: age, gender, marital status, educational/academic qualifications, professional situation and socio-economic status according to monthly income [12].

Academic qualifications were classified using the International Standard Classification of Education (ISCED), adopted by the United Nations Educational, Scientific and Cultural Organisation [19]. The selection of these variables was based on their applicability in previous studies and their relevance to the current research context.

## Dependent variables

The dependent variables analysed in this study included functional disability, assessed by the WHODAS 2.0, depressive symptoms, measured by the PHQ-9, self-reported satisfaction with

life, assessed by the self-perceived satisfaction with life score, and satisfaction with social support, assessed by the Pais-Ribeiro Satisfaction with Social Support Scale. These variables were selected to provide a comprehensive understanding of the participants' physical, mental and social well-being, and their relationship with sociodemographic factors.

## Data analysis

Data analysis was performed using IBM SPSS Statistics, version 29. Initially, a descriptive analysis was carried out to examine the distribution of the dependent variables according to the participants' independent variables. In order to identify statistically significant differences between the means of the variables in different groups, Student's t and one-factor ANOVA were applied to identify differences between groups. Statistical significance was set at $p < 0.05$.

## Ethical considerations

The study protocol was approved by the Ethics Committee of the University of Évora (Document 22093) and by the Scientific Council. It was also approved by the Ethics Committee of the Local Health Unit of Baixo Alentejo - ULSBA (Minute No. 1 of 2023), approved by the Board of Directors on 11/01/2023 (Ata N. 2, Point 4.1).

All participants were informed of the aims of the study, which were presented in writing in a written statement of free and informed consent in the first section of the self-administered questionnaire. Written informed consent to the study was obtained prior to completing the questionnaire. Participants who did not consent to their participation were not given access to the content of the questionnaire. Participant confidentiality and data anonymisation were strictly maintained.

## Results

### Characteristics of middle-aged adults

The study sample consisted of 698 middle-aged adults aged between 55 and 64 years. All participants were registered at health centres within the Baixo Alentejo region. Table 1 presents the sociodemographic characteristics of the sample. Among the participants, 52.4% were aged 55–59, and 47.6% were aged 60–64. The mean age was 59.2 ± 2.9 years. Women constituted 63.9% of the sample, indicating a gender imbalance.

The majority of participants were married (62.6%), followed by those in common-law marriages (12.2%), divorced individuals (11.0%), single individuals (8.9%), and widowed participants (5.3%). Educational qualifications varied: 28.8% had primary education, 39.4% had secondary education, and 31.8% had higher education. Thus, 71.2% of participants had completed at least secondary education.

In terms of professional status, 79.8% were employed, 10.2% were self-employed, and 10.0% were unemployed. Regarding self-perceived financial status, 39.4% reported no financial difficulties, 49.6% reported some difficulty, and 11.0% reported significant financial challenges.

**Physical dimension (Disability/functionality assessment).** The relationship between sociodemographic variables (age, gender, marital status, academic qualifications, employment status and self-perception of monthly income) and the average WHODAS 2.0 PT-12 score was explored, as well as the impact of these variables on the individuals' daily activities (the data analysed includes the number of days with difficulties, the number of days unable to work or carry out usual activities, and the number of days with a reduction in these activities, without counting the days when they were unable).

Table 1. Socio-demographic characteristics of middle-aged adults (N = 698).

| Variables | | $n_i$ | % |
|---|---|---|---|
| Age group in years | [55; 60] | 366 | 52.4 |
| | [60; 64] | 332 | 47.6 |
| Sex | Male | 252 | 36.1 |
| | Female | 446 | 63.9 |
| Marital status | Single | 62 | 8.9 |
| | Married | 437 | 62.6 |
| | Common-law marriage | 85 | 12.2 |
| | Divorced | 77 | 11.0 |
| | Widowed | 37 | 5.3 |
| Academic qualifications | Primary | 201 | 28.8 |
| | Secondary | 275 | 39.4 |
| | Higher | 222 | 31.8 |
| Professional status | Unemployed | 70 | 10.0 |
| | Employed | 557 | 79.8 |
| | Self-employed | 71 | 10.2 |
| Self-perception of monthly income | Reaches the end of the month without difficulty | 275 | 39.4 |
| | Reaches the end of the month with some difficulty | 346 | 49.6 |
| | Reaches the end of the month with great difficulty | 77 | 11.0 |

The participants had an average WHODAS 2.0 PT-12 score, on a scale of 12 to 60 points, of 18.3 ± 7.2 (95% Confidence Interval [CI]: [17.8, 18.8]), with a minimum of 12 points and a maximum of 53. The 25th percentile is at 12 points and the 75th percentile at 22. There were 25.5 per cent ($n_i$ = 178) of participants who reported no disability (12 points in WHODAS 2.0, 25th percentile) and 26.6 per cent of participants with values equal to or greater than 22 points (75th percentile), corresponding to participants with more disability. Table 2 shows the distribution of mean WHODAS 2.0 scores according to the sociodemographic characteristics of middle-aged adults and the respective statistical tests.

The mean score on the WHODAS 2.0 PT-12, which measures disability/functionality, was slightly higher in the 60–64 age group (18.8 ± 7.6) compared to the 55–59 age group (17.8 ± 6.7), with a marginal value of statistical significance (p = 0.053). This result suggests an almost significant tendency for functional disability to increase (functional decline) with age. The average number of days with difficulties was also higher in participants aged 60–64 (8.9 days) compared to those aged 55–59, although this difference did not reach statistical significance (p = 0.111). There were no statistically significant differences in the average number of days the participants were unable to work or carry out their usual activities due to the disability (p = 0.239), although the values were higher in the older age group. However, there was a statistically significant difference in the number of days with reduced usual or work activities, with participants aged 60–64 showing higher values (p = 0.033).

Women had significantly higher WHODAS 2.0 scores (18.9 versus 17.2; p = 0.001) compared to men. They also reported more days with difficulties (9.1 versus 6.6; p = 0.006) and more days with reduced usual activities or work (3.8 versus 2.1; p = 0.002). However, there were no significant differences in the number of days unable to work or carry out normal activities (p = 0.239), although women reported a higher average.

No statistically significant differences were found between the different marital statuses in relation to the variables analysed. However, widowers have the highest WHODAS 2.0 scores

**Table 2. Distribution of mean WHODAS 2.0 PT12 scores according to sociodemographic characteristics of middle-aged adults (N = 698).**

| Variables | | $n_i$ | % | WHODAS 2.0 score [12;60] | | Number of days with difficulties [0;30] | | Number of days unable to work or carry out normal activities [0;30] | | Number of days with reduced habitual or work activities [0;30] | |
|---|---|---|---|---|---|---|---|---|---|---|---|
| | | | | Mean (SD) | p | Mean (SD) | p | Mean (SD) | p | Mean (SD) | p |
| Age group in years | [55; 60] | 366 | 52.4 | 17.8 (6.7) | .053† | 7.5 (11.1) | .111† | 1.8 (5.8) | .055† | 2.6 (6.5) | .033† |
| | [60; 64] | 332 | 47.6 | 18.8 (7.6) | | 8.9 (11.9) | | 2.8 (7.7) | | 3.8 (8.6) | |
| Sex | Male | 252 | 36.1 | 17.2 (6.7) | **.001†** | 6.6 (10.8) | .006† | 1.9 (6.5) | .239† | 2.1 (6.5) | .002† |
| | Female | 446 | 63.9 | 18.9 (7.3) | | 9.1 (11.8) | | 2.5 (6.9) | | 3.8 (8.1) | |
| Marital status | Single | 62 | 8.9 | 18.6 (7.5) | .224 Ω | 7.9 (11.4) | .934 Ω | 2.8 (7.9) | .928 Ω | 4.0 (9.1) | .759 Ω |
| | Married | 437 | 62.6 | 18.0 (6.8) | | 8.1 (11.4) | | 2.2 (6.6) | | 3.0 (7.3) | |
| | Common-law marriage | 85 | 12.2 | 18.5 (7.5) | | 8.3 (11.8) | | 2.7 (7.4) | | 3.4 (7.9) | |
| | Divorced | 77 | 11.0 | 18.3 (7.6) | | 9.3 (12.3) | | 2.1 (6.6) | | 3.6 (8.1) | |
| | Widowed | 37 | 5.3 | 20.8 (8.5) | | 7.9 (11.1) | | 2.3 (5.2) | | 2.2 (6.2) | |
| Academic qualifications | Primary | 201 | 28.8 | 19.3 (7.5) | .003 Ω | 9.3 (12.2) | .056 Ω | 3.6 (8.5) | .002 Ω | 4.1 (9.0) | .096 Ω |
| | Secondary | 275 | 39.4 | 18.6 (7.3) | | 8.5 (11.7) | | 2.1 (6.5) | | 2.6 (7.0) | |
| | Higher | 222 | 31.8 | 17.0 (6.4) | | 6.8 (10.4) | | 1.3 (4.8) | | 2.9 (6.9) | |
| Professional status | Unemployed | 70 | 10.0 | 21.6 (9.0) | <.001 Ω | 13.5 (13.8) | <.001 Ω | 4.8 (9.5) | .001 Ω | 5.1 (9.8) | .064 Ω |
| | Employed | 557 | 79.8 | 18.0 (6.8) | | 7.7 (11.0) | | 1.9 (6.0) | | 2.9 (7.1) | |
| | Self-employed | 71 | 10.2 | 17.3 (6.8) | | 6.7 (11.3) | | 3.2 (8.5) | | 3.3 (8.7) | |
| Self-perception of monthly income | Reaches the end of the month without difficulty | 275 | 39.4 | 16.5 (5.8) | <.001 Ω | 6.4 (10.5) | <.001 Ω | 1.8 (6.3) | .015 Ω | 2.6 (7.2) | .034 Ω |
| | Reaches the end of the month with some difficulty | 346 | 49.6 | 18.7 (7.2) | | 7.9 (10.9) | | 2.2 (6.5) | | 3.2 (7.2) | |
| | Reaches the end of the month with great difficulty | 77 | 11.0 | 22.9 (8.6) | | 16.1 (14.1) | | 4.3 (9.2) | | 5.1 (10.1) | |

Legend: SD- Standard Deviation;

†- t-Student;

Ω- ANOVA.

and a relatively low number of days with difficulties and days unable to work. Divorced people were found to have a higher average WHODAS 2.0 score (20.8 ± 8.5), while single people reported the highest number of days unable to work or perform usual activities (2.8 ± 7.9) and reduced usual or work activities (4.0 ± 9.1).

Individuals with higher education had significantly lower WHODAS 2.0 scores ($p = 0.003$), reported fewer days with difficulties ($p = .056$), and fewer days unable to work ($p = 0.002$) compared to participants with basic education. Participants with basic education reported the worst results in all categories.

The unemployed show worse conditions in all the variables analysed. They have significantly higher WHODAS 2.0 scores ($p < 0.001$), more days with difficulties, more days unable to work, and more days with a reduction in usual activities (all with $p < 0.001$) compared to employees, especially salaried employees.

Analysing self-perception of monthly income indicates that the greater the perceived difficulty in managing income, the higher the WHODAS 2.0 score ($p < 0.001$) and the greater the impact on daily activities. Individuals who reported having great difficulty reaching the end of the month had the worst averages in all the variables analysed.

The highest average number of days (consecutive or not) in the month prior to answering WHODAS 2.0 refers to participants who had difficulties carrying out identified activities due to their health condition. As can be seen in Table 3, in one month, the average time spent by participants who reported having difficulties and those who reported being unable to carry out their usual activities or work, represents around 27.3 per cent of days (slightly more than a week). The total impossibility of carrying out normal activities or even working, due to the health condition, slightly exceeds two days on average, indicating that this is the least frequent situation. In each of the three situations indicated, the high variability of the number of days in relation to the average stands out, so the value of this statistical measure should be interpreted with some caution.

**Mental dimension: Presence of depressive symptoms and satisfaction with life.** The mental dimension is characterised by the average value of the PHQ9 score (which measures the presence and severity of depressive symptoms), the degree of difficulty that the problems reported on the PHQ9 scale cause at work, in looking after things at home or socialising with other people, and the score on the life satisfaction scale. The PHQ-9 scores reflect the severity of the reported depressive symptoms, while the life satisfaction index, assessed on a scale of 0 to 10, measures the subjective perception of well-being.

The mean overall score of the PHQ9 was 4.6 ± 5.1 (95% CI: [4.3, 5.0]), on a scale of 0 to 27, with a minimum and maximum of 0 and 26 points respectively; the 25th and 75th percentiles were 0 and 2.

Table 4 shows the distribution of participants according to depressive symptoms (number and percentage of participants). It can be seen that the majority of participants (63.0%) had no depressive symptoms, 21.5% reported mild symptoms and the lowest percentage (1.9%) reported severe depressive symptoms. Major depressive symptoms were reported by 15.5% of the participants.

Regarding the distribution of participants according to the level of difficulty in performing their job, looking after their own things at home or socialising with other people, caused by the depressive symptoms reported on the PHQ9 scale, it was observed that 44.8% of participants reported no difficulty. The largest proportion of participants (52.3%) reported little or a lot of difficulty, while a small proportion of 2.9% reported facing extreme difficulties.

In the self-assessment of life satisfaction, the average score was 7.6 ± 2.0 (95% CI: 7.4–7.7), on a scale of 0 to 10, with minimum and maximum values of 0 and 10 respectively. The 25th and 75th percentiles were 7 and 9. Of the participants, 75.4% scored 7 or more, indicating satisfaction with life, while 24.6% scored below 7, indicating dissatisfaction.

Table 5 shows the distribution of mean PHQ9 scores and life satisfaction scores according to the sociodemographic characteristics of middle-aged adults. Analysing by age group reveals that participants aged between 60 and 64 had a higher average PHQ9 score (5.0 ± 5.6) compared to those aged between 55 and 59 (4.3 ± 4.4). However, this difference did not reach

**Table 3. Average number of days with disability in the last 30 days (N = 698).**

| Number of days with: | Minimum | Maximum | mean (M) | 95% Confidence Interval | Standard Deviation (SD) | Coefficient of variation (CV) |
|---|---|---|---|---|---|---|
| Presence of difficulty in carrying out activities due to health condition | 0 | 30 | 8.2 | [7.3, 9.1] | 11.5 | 140.2% |
| Total inability to carry out usual activities or work due to health condition | 0 | 30 | 2.3 | [1.8, 2.8] | 6.8 | 295.7% |
| Decrease or reduction in usual activities or work due to health condition (not counting days of total impossibility) | 0 | 30 | 3.1 | [2.6, 3.7] | 7.6 | 245.2% |

statistical significance ($p$ = 0.093), suggesting only a tendency for depressive symptoms to increase with age. As for the life satisfaction score, there were no statistically significant differences between the age groups, with averages of 7.6 ± 1.9 for 55-59 year olds and 7.5 ± 2.1 for 60-64 year olds ($p$ = 0.614).

The results indicate that women had a significantly higher mean score on the PHQ9 (5.4 ± 5.2) compared to men (3.2 ± 4.4), with a statistically significant difference ($p$ < 0.001). On the other hand, men reported greater satisfaction with life, with an average of 7.9 ± 1.7, compared to 7.4 ± 2.1 in women ($p$ < 0.001), indicating an association between the gender variable and subjective well-being.

Although no statistically significant differences were found in the mean PHQ-9 score between the different marital statuses ($p$ = 0.208), widowers had the highest score (6.2 ± 5.5), suggesting a higher level of depressive symptoms. Satisfaction with life varied between the different marital statuses, with married people reporting the highest average satisfaction (7.7 ± 1.9) and divorced people the lowest (7.0 ± 2.2), although the difference did not reach statistical significance ($p$ = 0.066).

PHQ-9 scores varied according to level of education, with participants with higher education showing a lower average score (4.0 ± 4.6) compared to participants with primary education (5.0 ± 5.4), with a marginally significant difference ($p$ = 0.087). With regard to life satisfaction, there was a significant difference ($p$ = 0.043), with participants with higher education reporting greater satisfaction (7.9 ± 1.6).

Employment status had a significant influence on PHQ9 scores, with the unemployed showing the highest scores (6.5 ± 6.0) compared to the employed (4.4 ± 4.9; $p$ = 0.004). The employed also reported greater satisfaction with life (7.6 ± 1.9), in contrast to the unemployed (6.9 ± 2.7; $p$ = 0.005).

Analysing self-perception of monthly income revealed a strong association between perceived financial difficulty and PHQ9 scores. Individuals who reported reaching the end of the month with great difficulty had the highest average PHQ9 score (7.4 ± 6.2), significantly higher than those who did not report financial difficulties (3.5 ± 4.1; $p$ < 0.001). Similarly, life satisfaction was significantly higher among individuals without financial difficulties (8.1 ± 1.7) compared to those who reported major difficulties (6.7 ± 2.2) ($p$ < 0.001).

**Social dimension: Satisfaction with social support.** The social dimension is characterised by the average score obtained on the Satisfaction with Social Support Scale. The score on the scale ranges from 15 to 75 points and reflects the level of satisfaction with social support perceived by individuals. On the 15–75 point scale, the higher the score, the greater the

**Table 4. Distribution of participants according to depressive symptoms (N = 698).**

| Depressive conditions | | ni | (%) |
|---|---|---|---|
| Depressive symptoms (depression severity[a]) | Absence | 440 | 63.0 |
| | Mild | 150 | 21.5 |
| | Moderate | 65 | 9.3 |
| | Moderately severe | 30 | 4.3 |
| | Severe | 13 | 1.9 |
| Major depressive symptom[b] | No | 590 | 84.5 |
| | Yes | 108 | 15.5 |

Legend:

[a]depression severity category (PHQ9 score): absence (0–4), mild (5–9), moderate (10–14), moderately severe (15–19), severe (20–27);

[b]Major depressive symptom cut-off point: yes (PHQ9 score10), no (PHQ9 score < 10).

**Table 5. Distribution of mean PHQ9 scores and life satisfaction score according to sociodemographic characteristics of middle-aged adults (N = 698).**

| Variables | | $n_i$ | % | Mean PHQ-9 score | | Mean life satisfaction score | |
|---|---|---|---|---|---|---|---|
| | | | | Mean (SD) [0; 27] | p | Mean (SD) [0; 10] | p |
| Age group in years | [55; 60] | 366 | 52.4 | 4.3 (4.4) | .093† | 7.6 (1.9) | .614† |
| | [60; 64] | 332 | 47.6 | 5.0 (5.6) | | 7.5 (2.1) | |
| Sex | Male | 252 | 36.1 | 3.2 (4.4) | <.001† | 7.9 (1.7) | <.001† |
| | Female | 446 | 63.9 | 5.4 (5.2) | | 7.4 (2.1) | |
| Marital status | Single | 62 | 8.9 | 3.8 (4.6) | .208Ω | 7.5 (2.1) | .066Ω |
| | Married | 437 | 62.6 | 4.5 (4.9) | | 7.7 (1.9) | |
| | Common-law marriage | 85 | 12.2 | 4.9 (5.5) | | 7.4 (1.9) | |
| | Divorced | 77 | 11.0 | 4.7 (5.3) | | 7.0 (2.2) | |
| | Widowed | 37 | 5.3 | 6.2 (5.5) | | 7.5 (2.2) | |
| Academic qualifications | Primary | 201 | 28.8 | 5.0 (5.4) | .087Ω | 7.4 (2.3) | .043Ω |
| | Secondary | 275 | 39.4 | 4.9 (5.1) | | 7.5 (2.0) | |
| | Higher | 222 | 31.8 | 4.0 (4.6) | | 7.9 (1.6) | |
| Professional status | Unemployed | 70 | 10.0 | 6.5 (6.0) | .004Ω | 6.9 (2.7) | .005Ω |
| | Employed | 557 | 79.8 | 4.4 (4.9) | | 7.6 (1.9) | |
| | Self-employed | 71 | 10.2 | 4.2 (5.3) | | 8.0 (2.0) | |
| Self-perception of monthly income | Reaches the end of the month without difficulty | 275 | 39.4 | 3.5 (4.1) | <.001Ω | 8.1 (1.7) | <.001Ω |
| | Reaches the end of the month with some difficulty | 346 | 49.6 | 4.9 (5.1) | | 7.3 (2.0) | |
| | Reaches the end of the month with great difficulty | 77 | 11.0 | 7.4 (6.2) | | 6.7 (2.2) | |

Legend: SD- Standard Deviation;

†- t-Student;

Ω - ANOVA.

satisfaction with social support. The average global score obtained by the participants is 57.9 ± 10.6 (95% CI: 57.1–58.6), with a minimum of 21 and a maximum of 75. The 25th percentile is 51 points and the 75th percentile is 66 points.

Table 6 shows the distribution of the mean scores on the Satisfaction with Social Support Scale according to the sociodemographic characteristics of middle-aged adults. The average scores on the Satisfaction with Social Support Scale do not vary significantly between the different age groups. Participants aged between 55 and 59 showed an average score of 58.0 ± 10.3, while those aged between 60 and 64 showed a slightly lower average of 57.7 ± 11.0 ($p = 0.658$). These results indicate that satisfaction with social support is consistent between the age groups analysed.

Analysing the differences between the sexes reveals that men tend to be more satisfied with social support than women. The average score for men is 59.7 ± 10.4, significantly higher than for women, who have an average of 56.8 ± 10.6 ($p < 0.001$). This finding suggests a difference between the gender variable in the perception of social support, indicating that men feel more satisfied.

Although no statistically significant differences were found in the mean score of the satisfaction with social support scale between the different marital statuses of the participants ($p = 0.154$), the results show that divorced individuals have the lowest mean score (55.3 ± 12.1), while single and married individuals have similar mean scores (58.4 ± 9.4 and 58.4 ± 10.6, respectively).

**Table 6. Distribution of mean scores on the satisfaction with social support scale according to sociodemographic characteristics of middle-aged adults (N = 698).**

| Variables | | $n_i$ | % | Satisfaction with Social Support Scale mean score | |
|---|---|---|---|---|---|
| | | | | Mean (SD) [15; 75] | p |
| Age group in years | [55; 60] | 366 | 52.4 | 58.0 (10.3) | .658† |
| | [60; 64] | 332 | 47.6 | 57.7 (11.0) | |
| Sex | Male | 252 | 36.1 | 59.7 (10.4) | **<.001†** |
| | Female | 446 | 63.9 | 56.8 (10.6) | |
| Marital status | Single | 62 | 8.9 | 58.4 (9.4) | .154 Ω |
| | Married | 437 | 62.6 | 58.4 (10.6) | |
| | Common-law marriage | 85 | 12.2 | 57.5 (10.6) | |
| | Divorced | 77 | 11.0 | 55.3 (12.1) | |
| | Widowed | 37 | 5.3 | 56.5 (8.7) | |
| Academic qualifications | Primary | 201 | 28.8 | 57.9 (10.3) | .116 Ω |
| | Secondary | 275 | 39.4 | 56.9 (10.4) | |
| | Higher | 222 | 31.8 | 58.9 (11.1) | |
| Professional status | Unemployed | 70 | 10.0 | 54.6 (11.8) | **.004 Ω** |
| | Employed | 557 | 79.8 | 57.9 (10.4) | |
| | Self-employed | 71 | 10.2 | 60.4 (10.1) | |
| Self-perception of monthly income | Reaches the end of the month without difficulty | 275 | 39.4 | 60.7 (9.9) | **<.001 Ω** |
| | Reaches the end of the month with some difficulty | 346 | 49.6 | 56.6 (10.3) | |
| | Reaches the end of the month with great difficulty | 77 | 11.0 | 53.2 (11.8) | |

Legend: SD- Standard Deviation;

†- t-Student;

Ω - ANOVA.

Analysing educational qualifications reveals a non-significant trend towards greater satisfaction with social support among participants with higher levels of education. The average score is higher among individuals with higher education (58.9 ± 11.1) and lower among participants with secondary education (56.9 ± 10.4), with a marginally non-significant difference ($p$ = 0.116).

Professional status was significantly associated with satisfaction with social support (p = 0.004). The unemployed had the lowest average score (54.6 ± 11.8), while the self-employed reported the highest average (60.4 ± 10.1).

Self-perception of monthly income is strongly associated with satisfaction with social support ($p$ < 0.001). Participants who reported reaching the end of the month without difficulties had the highest mean score (60.7 ± 9.9), while those who reported major financial difficulties had the lowest score (53.2 ± 11.8).

## Discussion

This study characterised 698 middle-aged adults, aged between 55 and 64, registered at health centres in the Baixo Alentejo region. Analysing sociodemographics and physical, mental and social dimensions provided an in-depth view of the determinants of active and healthy ageing, offering a solid basis for future interventions and policies to promote quality of life in this age group, which is part of Portugal's oldest region.

This characterisation and analysis (at the same time physical, mental and social) not only illustrates the specific characteristics of the middle-aged population in the Baixo Alentejo, but

also highlights its uniqueness in a European region where the dynamics of ageing require a differentiated approach.

The results highlight the relationship between socio-demographic variables and functional difficulties, which are particularly relevant from the perspective of active ageing. The association observed between gender, employment status and self-perceived monthly income with physical disability, depressive symptoms and satisfaction with social support emphasises the need to consider these variables when developing strategies to support physical, mental and social well-being. This association reinforces the importance of considering these variables when creating strategies to support physical, mental and social well-being in middle-aged adults. Knowledge of sociodemographic factors is important in the study of the health of middle-aged populations, as it underpins the development of lifelong interventions on healthy ageing [20,21].

The ageing of the population in Portugal is a marked phenomenon, with 22% of the population aged 65 or over in 2019, a proportion that has doubled since 1971. Portugal is the fourth country in the world with the highest percentage of elderly people, after Japan (28%), Italy (23%) and Finland (22%) [1,22–24]. The Alentejo region (to which Baixo Alentejo belongs) is one of the oldest regions in Portugal, a trend that can also be observed in other parts of the world.

The findings highlight challenges common to ageing populations worldwide, including the association between lower educational attainment and greater functional disability; gender differences in mental health, particularly higher depressive symptoms in women; and the impact of lower socioeconomic status on physical limitations, depressive symptoms and dissatisfaction with social support. These findings highlight the need for lifelong learning initiatives to mitigate physical decline in later life [2,12,25–27]. These issues are not confined to specific regions, but are observed worldwide. In Southern Europe, East Asia and rural areas of the United States, lower socioeconomic status has been associated with increased physical limitations and depressive symptoms in older adults [12,28]. In addition, studies suggest that depressive symptoms may exacerbate functional disability, thereby exacerbating health inequalities within ageing populations [25,26,29,30].

Addressing these challenges requires integrated, transdisciplinary and multisectoral interventions that link health, economic and social support systems. Implementing policies that promote lifelong learning, strengthen social support networks and ensure economic stability are key steps in mitigating the adverse effects of ageing and promoting equitable health outcomes worldwide.

In the physical dimension, the assessment of functionality/disability by WHODAS 2.0 indicated that women and the unemployed have significantly higher levels of functional difficulty, affecting the capacity for daily activities and labour productivity. This finding alerts us to the need to personalise interventions, as it is essential to deal with physical frailty in older women [31,32,34], echoing the premise that strategies adapted to the unique context of the Baixo Alentejo are fundamental. Thus, the implementation of interventions geared towards professional and financial support may be essential to reducing functional difficulties among the most vulnerable groups. There is evidence of an association between physical activity and successful ageing in middle-aged and older adults, reinforcing that being physically active in middle and old age is beneficial for successful ageing [35–37].

Interventions to promote healthy ageing in order to reduce frailty are particularly relevant in middle age, as this is a stage in the development of life that is ideal for developing preventive interventions [34,35,38,39]. Health promotion interventions that include physical activity, such as walking and exercise, could help maintain functionality throughout the ageing process. Health promotion interventions for middle-aged adults can significantly improve healthy

lifestyles, protecting them from the negative effects of ageing [40]. The evidence synthesised by these authors supports the effectiveness of interventions that have contributed to positive changes at the biopsychosocial level, such as self-realisation, improved physical condition, behaviour that adheres to healthy lifestyles, increased quality of life and well-being [40]. This focus is consistent with the initial argument, which emphasises the urgency of adapting health policies to the reality of middle-aged adults in the regions most affected by ageing.

In the mental dimension, the results showed a statistically significant association between gender, occupational status and depressive symptoms. In particular, women and the unemployed reported higher scores on the PHQ-9, highlighting the influence of socio-economic factors on psychological well-being. These findings are consistent with research showing greater susceptibility to depressive states in groups with financial and employment instability, and reinforce the need for a multifactorial approach to the phenomenon of active ageing, taking into account the biopsychosocial influences on the mental health of middle-aged adults. Positive self-perceptions of income are associated with greater life satisfaction, reinforcing the importance of economic security for psychological well-being. Life satisfaction increases throughout life for those who are better off economically and who are more likely to meet their acquaintances [27,41,42]. Satisfaction with life decreases over time as the number of chronic illnesses increases, especially compared to those who were normal weight or depressed versus not depressed [43,44]. This evidence is similar to that of the present study, pointing to practical implications for health professionals and policymakers regarding ways to improve life satisfaction, indicating the need to pay more attention to vulnerable middle-aged adults. More research is needed to identify the mechanisms that may explain the changes and related factors in middle-aged adults, as observed in this study.

With regard to the social dimension, a significant association was found between economic difficulties and levels of satisfaction with social support. Participants with a positive self-perception of income tended to report greater satisfaction with social support, highlighting the importance of financial stability for the perception of social support. Existing literature suggests that economic security favours stronger support networks and a more enriching social experience. Financial stability is related to the quality of social interactions [45–47]. Social support is relevant to maintaining cognitive health in populations, which underpins the importance of social connections in middle age [48,49].

Although marital status did not show a statistically significant association with satisfaction with social support ($p$ = 0.154), it was observed that divorced participants had lower scores, suggesting that marital status can influence the perception of social support. These data are consistent with the literature which indicates that social support networks in married or partnered individuals can act as a protective factor [50]. Mohanty et al. (2021), in a study that analysed the association between socioeconomic variables and health variables in middle-aged and elderly people, highlighted the association between the physical proximity of family members and health conditions, in addition to economic conditions [51]. This indicates the need to explore this variable in future studies, especially considering the complexity of social dynamics in the various stages of life [33,52,53].

Physical, psychological and cognitive disorders are prevalent in populations with low socioeconomic status, but the association between this status and multimorbidity is unclear, especially outside of high-income countries. The study by Ni et al. (2023) assessed these associations in 33 countries of different income levels and concluded that, in the populations analysed, the odds of multimorbidity were more than ten times higher among individuals with low socioeconomic status [28]. Thus, the implementation of policies and programmes focused on equity is urgently needed to reduce social inequalities in health and achieve the Sustainable Development Goals. This study corroborates this statement.

In general, the results highlight the need for specific policies and interventions for the groups identified as most vulnerable, with emphasis on socio-economic, labour and psychological support. Promoting healthy ageing, which is a process of developing and maintaining functional capacity that enables well-being in old age [54], is crucial to responding to the growing number of ageing populations.

### Limitations and strengths

The study's large, representative sample enables generalisability to the population of middle-aged adults in Baixo Alentejo. However, the cross-sectional design limits causal inferences, highlighting the need for longitudinal studies to explore the long-term impacts of sociodemographic variables on ageing trajectories. Future research should also consider a more diverse sample to examine variations across different socioeconomic and cultural contexts.

### Future studies, policies and practice

The results highlight the strong link between socio-economic factors and mental wellbeing, and underline the significant impact of gender and employment status. Women and the unemployed reported higher levels of depressive symptoms, while men reported higher levels of life satisfaction. Economic stability was found to be positively associated with both mental well-being and life satisfaction, highlighting the crucial role of financial security in shaping perceptions of social support. These findings point to the need for policy interventions that address socio-economic inequalities. In particular, policies aimed at promoting economic stability and strengthening social support systems are essential for improving health outcomes, especially for vulnerable populations.

The results emphasise the importance of developing public policies and interventions that address the specific needs of middle-aged adults. Strategies should prioritise physical activity, economic inclusion, mental health support, and the strengthening of social networks. Community programmes, in particular, can play a pivotal role in mitigating the negative effects of ageing by fostering social connections and emotional support [55,56].

In addition, the development of specific interventions must include strategies that directly address the inequalities identified, personalised approaches that consider the specific needs of the most vulnerable individuals [12,31,55], and educational and motivational health promotion interventions that translate into health gains [40]. The relationship between financial stability and the quality of social support suggests that policies should be focussed on economic inclusion and improving living conditions, which can have a positive impact on the psychological and social well-being of middle-aged adults. The promotion of robust social support networks is crucial for mental health and quality of life, particularly in economically disadvantaged populations [56].

Finally, it is imperative that policymakers and health professionals take the evidence presented into consideration in order to develop strategies that adequately address the needs of middle-aged adults in the Baixo Alentejo. Collaboration between different sectors is vital to promote active, healthy and sustainable ageing, ensuring that all individuals have access to the support and resources necessary for a full and dignified life.

## Conclusion

To our knowledge, this study is a pioneer in adopting an integrative approach aimed at identifying factors related to well-being at this stage of life. It has contributed to filling an existing gap in the literature and providing a basis for future research that considers the specific context of Baixo Alentejo as a critical determinant of active and healthy ageing.

The results highlight the urgent need for specific policies and interventions to promote active and healthy ageing. The analysis carried out shows that financial stability, social support and gender are critical determinants for the functionality, mental health and social satisfaction of the group studied. The implementation of strategies adapted to the sociodemographic profile of each group is fundamental to ensuring an inclusive and effective approach in the public health system, thus promoting quality longevity.

The findings reinforce the importance of public policies that prioritise economic inclusion, in line with the World Health Organisation's Sustainable Development Goals. Targeted interventions will not only contribute to improving the quality of life of middle-aged adults, but will also promote a holistic and sustainable approach to active ageing, in line with the best practices observed in international contexts.

The physical, mental and social dimensions of active and healthy ageing are interdependent and crucial to the well-being of older adults. Physical health, ensured through the regular practice of physical activities, is fundamental for maintaining functionality and preventing chronic diseases, thus promoting autonomy. At the same time, mental health, which includes stress management and the prevention of depression, is favoured by robust social support and the creation of support networks, showing that positive social interactions can mitigate loneliness and increase life satisfaction. The social dimension of ageing involves participation in community activities, which not only enriches the lives of older people, but also strengthens self-esteem and a sense of belonging. Therefore, a holistic approach that integrates these three dimensions is vital for maximising quality of life and promoting active and healthy ageing.

Finally, it is imperative to recognise the limitations of this research, namely its cross-sectional nature, which does not allow definitive causal relationships to be inferred. Future studies, preferably of a longitudinal nature, could deepen our understanding of the dynamics of health and well-being over time, providing a solid basis for the development of effective policies and interventions that meet the specific needs of the ageing population. This set of conclusions not only reaffirms the relevance of this study, but also establishes guidelines for future research and interventions aimed at promoting active and healthy ageing in the Baixo Alentejo region of Portugal.

## Author contributions

**Conceptualization:** Eunice Santos, Lara Guedes Pinho, Adelaide Proença, Helena Arco.

**Formal analysis:** Eunice Santos, Lara Guedes Pinho, Adelaide Proença, Helena Arco.

**Investigation:** Eunice Santos, Lara Guedes Pinho, Adelaide Proença, Helena Arco.

**Methodology:** Eunice Santos, Lara Guedes Pinho, Adelaide Proença, Helena Arco.

**Writing – original draft:** Eunice Santos, Lara Guedes Pinho, Adelaide Proença, Helena Arco.

**Writing – review & editing:** Eunice Santos, Lara Guedes Pinho, Adelaide Proença, Helena Arco.

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
