## [Decision Letter · Decision Letter 0]

29 Dec 2024

PONE-D-24-56736Exploring the Physical, Mental, and Social Dimensions of Middle-Aged Adults for Active and Healthy Aging: A Cross-Sectional StudyPLOS ONE

Dear Dr. Santos,

Thank you for submitting your manuscript to PLOS ONE. After careful consideration, we feel that it has merit but does not fully meet PLOS ONE’s publication criteria as it currently stands. Therefore, we invite you to submit a revised version of the manuscript that addresses the points raised during the review process.

We look forward to receiving your revised manuscript.

Kind regards,

André Ramalho, PhD

Academic Editor

PLOS ONE

Journal requirements: When submitting your revision, we need you to address these additional requirements. 1. Please ensure that your manuscript meets PLOS ONE's style requirements, including those for file naming. The PLOS ONE style templates can be found at https://journals.plos.org/plosone/s/file?id=wjVg/PLOSOne_formatting_sample_main_body.pdf and https://journals.plos.org/plosone/s/file?id=ba62/PLOSOne_formatting_sample_title_authors_affiliations.pdf.

Reviewers' comments:

Reviewer's Responses to Questions

**Comments to the Author**

1. Is the manuscript technically sound, and do the data support the conclusions?

Reviewer #1: Yes

Reviewer #2: Yes

2. Has the statistical analysis been performed appropriately and rigorously? 

Reviewer #1: I Don't Know

Reviewer #2: Yes

3. Have the authors made all data underlying the findings in their manuscript fully available?

Reviewer #1: Yes

Reviewer #2: Yes

4. Is the manuscript presented in an intelligible fashion and written in standard English?

Reviewer #1: Yes

Reviewer #2: Yes

5. Review Comments to the Author

Reviewer #1: STRENGTHS

a) The research is well formulated.

b) Scientific rigor is very evident.

c) Research methodology is good.

SUGGESTIONS:

a) Line 83. Please justify the assertion that there is a lack of integrative research on the topic by adding more citations as evidence to support this assertion. It is vital to justify the foundation of the research. Thank you.

b) The authors assert that Baixo Alentejo as one of the regions in Portugal with the highest ageing rates. It would be supportive for the article if the authors could justify choosing Baixo Alentejo above these other regions. May I please suggest adding more information to justify the choice of Baixo Alentejo as the research site? Thank you.

c) Were any regions in the ULSBA region left out of the survey. If yes, please justify the exclusion of these regions. Thank you.

d) In a lot of the literature and in various nations, the age of 60 is considered the beginning of old age. How do the authors justify the participants over this age as middle-aged and not older adults. Please clarify this point. Thank you.

e) In the participant exclusion criteria the authors exclude participants who cannot communicate in Portuguese. Would the authors please clarify who these participants are? Are they immigrants, migrant workers, spouses of Portuguese citizens from other countries etc.? A clarification from the authors is greatly appreciated. Thank you.

f) Line 442. How is this study unique from other parts of Europe? Kindly expound on this point. Thank you.

g) Would the authors consider, despite the study site being specifically Baixo Alentejo, expounding more on the implications of the findings on other communities with ageing populations? This would extend the impact of the article and its findings. Thank you.

Reviewer #2: El manuscrito en su resumen menciona los hallazgos, instrumentos, y metodología, se alinea a la literatura existente en relación con el envejecimiento, cita de manera clara los instrumentos utilizados. Se recomienda fortalecer la investigación incorporando estadísticas sobre el envejecimiento en la población de Portugal, aclarar en la metodología la selección de los participantes con la muestra representativa. Con respecto a los resultados detallar la interpretación de los datos especialmente las implicaciones de mayor puntuación de discapacidad entre las personas divorciadas y solteras, fortalecer la información para las políticas públicas en base a los resultados.

6. PLOS authors have the option to publish the peer review history of their article (what does this mean? ). If published, this will include your full peer review and any attached files.

**Do you want your identity to be public for this peer review?** For information about this choice, including consent withdrawal, please see our Privacy Policy .

Reviewer #1: No

Reviewer #2: No

---

## [Author Response · Author response to Decision Letter 1]

20 Jan 2025

Dear Reviewer 1

We would like to express our sincere gratitude for your thoughtful and constructive feedback on our manuscript. Your comments have been invaluable in improving the quality of our work, and we deeply appreciate the time and effort you have dedicated to reviewing our submission. Below, we provide our detailed responses to each of your points.

a) Justification of lack of integrative research on the topic (Line 83): Thank you for your suggestion to provide additional citations to support the assertion that there is a lack of integrative research on the physical, mental, and social well-being of middle-aged adults in Portugal. In response, we have reformulated the relevant section of the manuscript and included more references to better substantiate this claim. The revised text now reads as follows:

"Although global studies on ageing are extensive, there is a lack of integrative research addressing the physical, mental, and social well-being (three dimensions simultaneously) of middle-aged adults in Portugal, particularly in the Baixo Alentejo, where demographic changes and socio-economic challenges require targeted attention [1–4]. Existing studies mainly focus on the elderly population and rarely explore the intersectoral dimensions of well-being among middle-aged adults in this under-researched region [2,3,5,6]. This gap limits the development of policies and interventions tailored to regional needs. Given that middle age represents a pivotal window for preventive action, addressing physical, mental, and social dimensions during this stage is essential for promoting active ageing strategies [14].”

We hope this revised version meets your expectations and clarifies the research gap. Please do let us know if further adjustments are needed.

b) Justification for choosing Baixo Alentejo: We appreciate your comment and understand the importance of providing a stronger justification for selecting Baixo Alentejo as the study region. We have expanded on the demographic and socio-economic context of the region, including specific population trends and the unique challenges it faces. The revised text now reads as follows:

"The Alentejo region, which includes the Baixo Alentejo, could see a significant decline in population between 2000 and 2050, with the resident population falling by 46.1 per cent [1]. In particular, the young population is expected to fall by 70.4 per cent and the number of pre-school children by 72.9 per cent [1]. This makes it the Portuguese region with the highest ageing rate (189.2 per cent) [1–3]. Baixo Alentejo (NUTS III) is a predominantly rural region with the largest surface area in the country (8,544.6 km²) and the lowest population density (14.7 inhabitants/km²) [3–5]. Demographic challenges are exacerbated by long distances between towns, which can exceed 120 km, and a limited and inefficient public transport network [3,6]. Geographical dispersion and transport barriers make it difficult to provide effective health care, especially for the most dependent [2,6]. In this context, the ageing of the population in the Baixo Alentejo is a critical challenge, highlighting the urgent need for strategies to mitigate the socio-economic and public health impact of this phenomenon [2,3,7].”

We hope this additional context provides a clearer rationale for the selection of Baixo Alentejo. Should you require further details or clarification, we are happy to provide them.

c) Exclusion of regions in the ULSBA: Thank you for your question. We would like to confirm that all regions within the ULSBA catchment area were included in the study, and there were no exclusions. We trust this answers your query, but if further clarification is needed, please do not hesitate to ask. The revised text now reads as follows: “All regions in the ULSBA catchment area were included in the study.”

d) Justification for considering participants over 60 years old as middle-aged: We appreciate your query regarding the classification of participants over 60 years old as middle-aged. In Portugal, the age of 65 is generally considered the threshold for "elderly," as defined by national social and health policies, including the National Programme for Older People. The WHO and other national institutions commonly define middle age as ranging from 45 to 64 years. In line with these definitions, we have considered individuals aged 55 to 64 as middle-aged in our study.

According to Medical Subject Headings, a middle-aged adult is an adult aged 45–64 years. In Portugal, the definition of "elderly" generally applies to individuals aged 65 and over, as established by various social and public health policies, including the National Programme for Older People of the Directorate-General of Health. The National Statistics Institute also considers individuals 65 and older as elderly in national statistics. As for middle age, although the concept is flexible, it is typically understood in Portugal to span from 45 to 64 years of age. This phase represents a transition between youth and old age, marked by important physical and social changes. The WHO defines middle age as generally ranging from 45 to 59 years, although it may extend to 64 in some contexts. The Observatory of Inequalities also adopts the age range of 45 to 64 years for characterising middle age.

We hope this clarifies the reasoning behind our classification of middle age in the study.

e) Exclusion of participants who cannot communicate in Portuguese: Thank you for raising this point. The exclusion criteria were applied to participants who were unable to read, understand, or respond in Portuguese, as the study's questionnaire was written in Portuguese. This criterion ensured that participants could fully comprehend the content and provide accurate responses. The relevant clarification has been added to the manuscript:

" As the questionnaire was written in Portuguese and was self-administered, participants were required to be able to read and understand the language, which was essential for understanding the content and providing accurate responses."

We hope this explanation addresses your concern.

f) Uniqueness of the study compared to other parts of Europe: We appreciate your request for clarification on the uniqueness of this study. We have expanded on this point in the manuscript, emphasising that the simultaneous analysis of physical, mental, and social well-being in a specific region of Southern Europe is what sets this study apart. The revised section reads as follows:

"This characterisation and analysis (at the same time physical, mental, and social) not only illustrates the specific characteristics of the middle-aged population in the Baixo Alentejo, but also highlights its uniqueness in a European region where the dynamics of ageing require a differentiated approach."

We hope this revision provides a clearer explanation of the study's distinctive nature.

g) Implications of findings on other ageing communities: We appreciate your suggestion to expand on the implications of our findings for other communities with ageing populations. We have revised the manuscript to include a broader perspective, highlighting the global relevance of our findings and their implications for other ageing populations. The updated text reads as follows:

"The ageing of the population in Portugal is a marked phenomenon, with 22% of the population aged 65 or over in 2019, a proportion that has doubled since 1971. Portugal is the fourth country in the world with the highest percentage of elderly people, after Japan (28%), Italy (23%) and Finland (22%) [1,23–25]. The Alentejo region (to which Baixo Alentejo belongs) is one of the oldest regions in Portugal, a trend that can also be observed in other parts of the world. The findings highlight challenges common to ageing populations worldwide, including the association between lower educational attainment and greater functional disability; gender differences in mental health, particularly higher depressive symptoms in women; and the impact of lower socioeconomic status on physical limitations, depressive symptoms and dissatisfaction with social support. These findings highlight the need for lifelong learning initiatives to mitigate physical decline in later life [2,26–29]. These issues are not confined to specific regions, but are observed worldwide. In Southern Europe, East Asia and rural areas of the United States, lower socioeconomic status has been associated with increased physical limitations and depressive symptoms in older adults [13,30,30]. In addition, studies suggest that depressive symptoms may exacerbate functional disability, thereby exacerbating health inequalities within ageing populations [26,27,31,32]. Addressing these challenges requires integrated, transdisciplinary and multisectoral interventions that link health, economic and social support systems. Implementing policies that promote lifelong learning, strengthen social support networks and ensure economic stability are key steps in mitigating the adverse effects of ageing and promoting equitable health outcomes worldwide."

We trust that this additional information provides a more comprehensive perspective on the broader implications of our findings.

Once again, thank you for your valuable suggestions and observations, which have undoubtedly enhanced the quality of the manuscript. Should you require any further clarification or additional information, please do not hesitate to contact us. We remain at your disposal for any further inquiries.

Yours sincerely,

Eunice Santos

Dear Reviewer #2,

We would like to thank you for your valuable feedback and the insightful suggestions that you provided. Your comments were instrumental in improving the clarity and depth of the manuscript. Below, we address each of the points you raised and outline the revisions made in response.

a) Strengthening of the research with statistical data on ageing in Portugal: We appreciate your suggestion to incorporate statistics on ageing in the Portuguese population, particularly in the context of the Baixo Alentejo. In response, we have enhanced the introduction of the manuscript by including relevant demographic data on the region's population decline and ageing trends. This addition helps contextualise the importance of the study and the challenges faced by this region. The revised text is as follows:

"The Alentejo region, which includes the Baixo Alentejo, could see a significant decline in population between 2000 and 2050, with the resident population falling by 46.1 per cent [1]. In particular, the young population is expected to fall by 70.4 per cent and the number of pre-school children by 72.9 per cent [1]. This makes it the Portuguese region with the highest ageing rate (189.2 per cent) [1–3]. Baixo Alentejo (NUTS III) is a predominantly rural region with the largest surface area in the country (8,544.6 km²) and the lowest population density (14.7 inhabitants/km²) [3–5]. Demographic challenges are exacerbated by long distances between towns, which can exceed 120 km, and a limited and inefficient public transport network [3,6]. Geographical dispersion and transport barriers make it difficult to provide effective health care, especially for the most dependent [2,6]. In this context, the ageing of the population in the Baixo Alentejo is a critical challenge, highlighting the urgent need for strategies to mitigate the socio-economic and public health impact of this phenomenon [2,3,7].”

This inclusion provides the necessary demographic context and strengthens the manuscript by reinforcing the urgency of addressing the challenges posed by population ageing in the region.

b) Clarification of the methodology and participant selection: We have clarified the methodology regarding the participant selection process, as suggested. The revision aims to make the description of the participant recruitment process more transparent and specific. The revised text is as follows:

"Each person attending the health centre for a consultation was invited by the nurse to participate in the study. The nurse explained the aims of the study, and if the person agreed, they were given the questionnaire to complete at that time, with sufficient time allowed."

We believe this revision now clearly explains the participant selection process and how individuals were recruited for the study.

c) Interpretation of results, particularly with regard to higher disability scores among divorced and single individuals: Thank you for your valuable comment regarding the interpretation of the results, particularly the higher disability scores observed among divorced and single individuals. We have expanded the discussion to address this point in more detail. The updated text is as follows:

"The results highlight the strong link between socio-economic factors and mental well-being, and underline the significant impact of gender and employment status. Women and the unemployed reported higher levels of depressive symptoms, while men reported higher levels of life satisfaction. Economic stability was found to be positively associated with both mental well-being and life satisfaction, highlighting the crucial role of financial security in shaping perceptions of social support. These findings point to the need for policy interventions that address socio-economic inequalities. In particular, policies aimed at promoting economic stability and strengthening social support systems are essential for improving health outcomes, especially for vulnerable populations."

This revision provides a more detailed interpretation of the results, particularly in relation to the disability scores and the implications for policy development.

d) Strengthening the policy recommendations: We have taken your recommendation to reinforce the information for public policy based on the results of the study. The revised conclusion now includes a more direct focus on policy recommendations aimed at addressing the issues identified in the study. The updated text is as follows:

"Although global studies on ageing are extensive, there is a lack of integrative research addressing the physical, mental, and social well-being (three dimensions simultaneously) of middle-aged adults in Portugal, particularly in the Baixo Alentejo, where demographic changes and socio-economic challenges require targeted attention [1–4]. Existing studies mainly focus on the elderly population and rarely explore the intersectoral dimensions of well-being among middle-aged adults in this under-researched region [2,3,5,6]. This gap limits the development of policies and interventions tailored to regional needs. Given that middle age represents a pivotal window for preventive action, addressing physical, mental, and social dimensions during this stage is essential for promoting active ageing strategies [14].”

This addition clarifies the importance of focusing on middle-aged adults and the need for policies that specifically target this demographic in light of the challenges faced by the Baixo Alentejo region.

We truly appreciate your thoughtful suggestions, which have greatly improved the quality of the manuscript. Should you have any further comments or require additional clarifications, we are happy to address them.

Thank you once again for your time and constructive feedback.

Kind regards,

Eunice Santos

Dear Prof. Dr. André Ramalho,

PLOS ONE Editor

I hope this message finds you well. I would like to express my sincere gratitude to you and the reviewers for your thoughtful and constructive feedback on our manuscript titled "Exploring the Physical, Mental, and Social Dimensions of Middle-Aged Adults for Active and Healthy Aging: A Cross-Sectional Study" (PONE-D-24-56736). We greatly appreciate the time and effort invested in reviewing our work.

In response to the comments and suggestions provided, we have carefully revised the manuscript to address all points raised. We believe these revisions have strengthened the overall quality of the paper. Attached, please find the following files for your consideration:

1. A detailed Response to Reviewers document, outlining how each comment has been addressed and specifying the corresponding changes made in the manuscript.

2. A Revised Manuscript with Track Changes

---

## [Decision Letter · Decision Letter 1]

18 Feb 2025

Exploring the Physical, Mental, and Social Dimensions of Middle-Aged Adults for Active and Healthy Aging: A Cross-Sectional Study

PONE-D-24-56736R1

Dear Dr. Santos,

We’re pleased to inform you that your manuscript has been judged scientifically suitable for publication and will be formally accepted for publication once it meets all outstanding technical requirements.

Kind regards,

André Luis C Ramalho, PhD

Academic Editor

PLOS ONE

**Additional Editor Comments:**

Following the R1 round of peer review, the manuscript titled "Exploring the Physical, Mental, and Social Dimensions of Middle-Aged Adults for Active and Healthy Aging: A Cross-Sectional Study" has undergone a thorough and rigorous evaluation. The first reviewer provided a positive assessment and recommended acceptance. The second reviewer, who had previously suggested minor revisions, did not respond to the request for reassessment despite multiple reminders and was consequently uninvited from the review process.

A detailed review of the authors’ revisions confirms that all concerns raised by both reviewers in the initial round have been fully and thoughtfully addressed. The manuscript has been substantially improved through the integration of comprehensive statistical data on aging in Portugal, a clearer justification of the participant selection methodology, an expanded interpretation of key findings—particularly regarding the higher disability scores observed among divorced and single individuals—and a reinforced discussion on the study’s implications for public policy. These enhancements have strengthened the manuscript’s methodological rigor, analytical depth, and overall clarity.

Beyond these technical refinements, the study makes a meaningful contribution to both the scientific community and public health discourse by addressing a critical yet often overlooked stage in the aging continuum. By examining the physical, mental, and social well-being of middle-aged adults in Baixo Alentejo, a region facing significant demographic and socioeconomic challenges, the research generates valuable insights with direct implications for health policy and intervention strategies. The findings highlight the intricate relationship between economic stability, social support, and gender disparities in shaping aging trajectories, emphasizing the need for proactive and inclusive approaches that extend beyond traditional elderly care. Given its empirical robustness, contextual significance, and potential to inform evidence-based policies, this manuscript represents a valuable resource for researchers, policymakers, and healthcare professionals committed to advancing population health and promoting quality longevity.

Given the substantial improvements made and the manuscript’s strong alignment with the journal’s scientific and methodological standards, I am confident that the peer review process has been satisfactorily completed. With all points raised in the initial review fully addressed, the first reviewer’s endorsement, and the second reviewer’s feedback fully incorporated, I am pleased to proceed with its acceptance.

**Sincerely,**

Prof. Dr. André Ramalho

Reviewers' comments:

Reviewer's Responses to Questions

**Comments to the Author**

1. If the authors have adequately addressed your comments raised in a previous round of review and you feel that this manuscript is now acceptable for publication, you may indicate that here to bypass the “Comments to the Author” section, enter your conflict of interest statement in the “Confidential to Editor” section, and submit your "Accept" recommendation.

Reviewer #1: All comments have been addressed

2. Is the manuscript technically sound, and do the data support the conclusions?

Reviewer #1: Yes

3. Has the statistical analysis been performed appropriately and rigorously? 

Reviewer #1: I Don't Know

4. Have the authors made all data underlying the findings in their manuscript fully available?

Reviewer #1: Yes

5. Is the manuscript presented in an intelligible fashion and written in standard English?

Reviewer #1: Yes

6. Review Comments to the Author

Reviewer #1: Thank you for the revision of the article. You have addressed the issues I have raised and the article is more solid in nature. Thank you.

7. PLOS authors have the option to publish the peer review history of their article (what does this mean? ). If published, this will include your full peer review and any attached files.

**Do you want your identity to be public for this peer review?** For information about this choice, including consent withdrawal, please see our Privacy Policy .

Reviewer #1: No

---

## [Editor Report · Acceptance letter]

PONE-D-24-56736R1

PLOS ONE

Dear Dr. Santos,

I'm pleased to inform you that your manuscript has been deemed suitable for publication in PLOS ONE. Congratulations! Your manuscript is now being handed over to our production team.

Kind regards,

on behalf of

Prof. Dr. André Luis C Ramalho

Academic Editor

PLOS ONE
